# Phenomenological study on correlation between flow harmonics and mean transverse momentum in nuclear collisions

**Chunjian Zhang[1⋆], Jiangyong Jia[1,2] and Shengli Huang[1]**

**1** Department of Chemistry, Stony Brook University, Stony Brook, NY 11794, USA
**2** Physics Department, Brookhaven National Laboratory, Upton, NY 11976, USA

⋆ chun-jian.zhang@stonybrook.edu

## Abstract

To assess the properties of the quark-gluon plasma formed in nuclear collisions, the Pearson correlation coefficient between flow harmonics and mean transverse momentum, $\rho\left(v_n^2, [p_{\mathrm{T}}]\right)$, reflecting the overlapped geometry of colliding atomic nuclei, is measured. $\rho\left(v_2^2, [p_{\mathrm{T}}]\right)$ was found to be particularly sensitive to the quadrupole deformation of the nuclei. We study the influence of the nuclear quadrupole deformation on $\rho\left(v_n^2, [p_{\mathrm{T}}]\right)$ in $\mathrm{Au} + \mathrm{Au}$ and $\mathrm{U} + \mathrm{U}$ collisions at RHIC energy using AMPT transport model, and show that the $\rho\left(v_2^2, [p_{\mathrm{T}}]\right)$ is reduced by the quadrupole deformation $\beta_2$ and turns to change sign in ultra-central collisions (UCC).

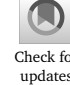

## 1 Introduction

A hot and dense phase of Quantum Chromodynamics (QCD) matter, so-called quark-gluon plasma (QGP) [1], is naturally created in the nuclear collisions studied at the Relativistic Heavy Ion Collider (RHIC) and the Large Hadron Collider (LHC). The QGP expands due to pressure gradients between the medium and the outside vacuum. During this expansion, the initial spatial anisotropies lead to final-state momentum anisotropies. The large azimuthal modulations in the final distributions of the produced particles are typically characterized as a Fourier series [2]: $\frac{dN}{d\phi} \propto 1 + 2\Sigma_{n=1}^{\infty} v_n \cos\left(n\left(\phi - \Phi_n\right)\right)$, where $v_n$ and $\Phi_n$ represent the magnitude and event-plane angle of the $n^{\mathrm{th}}$-order harmonic flow. Interestingly, the shape and orientation of the deformed nuclei characterized by the nuclear deformation, if taken into account for the generation of the initial state geometry, could result in the non-trivial behavior of final-state bulk observables [3–5].

Recently, the correlation between flow harmonics ($v_n$) and mean transverse momentum ($[p_T]$), $\rho(v_n\{2\}^2, [p_T])$ [6], was proposed to be sensitive to distinguish the nuclear deformation [7]. Of particular interest is a significantly negative correlation in central U+U collisions at STAR Collaboration [8] due to the quadrupole deformation $\beta_2$ where the $\beta_2$ values are obtained from the measured reduced electric transition probability $B(E_n)\uparrow$ via the standard formula $\beta_2 = \frac{4\pi}{3ZR_0^2}\sqrt{\frac{B(E2)\uparrow}{e^2}}$ [9]. Comparison with the $\beta_2$ scan in phenomenological study can explore the sensitivity of $\rho\left(v_n^2, [p_T]\right)$ to the fluctuations in the initial geometry arising from nuclei shape at a much shorter time scale ($\sim 10^{-24}$) in relativistic heavy-ion collisions. To further constrain the initial conditions and transport properties in hydrodynamic evolution, ATLAS [10,11] and ALICE Collaboration [12] have also reported this measurements in system scan $pp$, $p$+Pb, Xe+Xe, and Pb+Pb collisions.

In this proceeding, The influences of the nuclear quadrupole deformation on $\rho\left(v_n^2, [p_T]\right)$ in Au+Au and U+U collisions at $\sqrt{s_{NN}} = 200$ GeV are studied with the framework of AMPT transport model.

## 2 Analysis

The $\rho\left(v_n^2, [p_T]\right)$ is quantified by a three-particle correlator defined as:

$$\rho\left(v_n^2, [p_T]\right) = \frac{\langle v_n^2 \delta p_T \rangle}{\sqrt{\left\langle \left(\delta v_n^2\right)^2 \right\rangle \langle \delta p_T \delta p_T \rangle}} = \frac{\left\langle \frac{\sum_{i,j,k,i\neq j\neq k} e^{in\left(\phi_i - \phi_j\right)}\left(p_{T,k} - \langle[p_T]\rangle\right)}{\sum_{i,j,k,i\neq j\neq k}} \right\rangle}{\sqrt{\left(\langle v_n^4 \rangle - \langle v_n^2 \rangle^2\right)\left\langle \frac{\sum_{i,j,i\neq j}\left(p_{T,i} - \langle[p_T]\rangle\right)\left(p_{T,j} - \langle[p_T]\rangle\right)}{\sum_{i,j,i\neq j}} \right\rangle}} \,, \quad (1)$$

where the indices $i$, $j$ and $k$ loop over distinct particles to account for all unique triplets, and the $\langle\rangle$ denotes average over events. In this analysis, we use all particles within $|\eta| < 2$ and $0.2 < p_T < 2$ GeV/c for the benefit of statistical precision and the centrality is defined using the number of participants $N_{part}$. We use the AMPT model v2.26t5 [13] with string-melting mode and partonic cross section of 3.0 mb, which we check reasonably reproduce Au+Au flow data at RHIC. The systematic study of short-range "non-flow" effect from resonance-decays, jets and dijets was checked in Ref. [14].

## 3 Result

Figures 1 shows the Pearson correlation coefficients $\rho\left(v_n^2, [p_T]\right)$ for n=2 and 3 calculated in standard method using final-state hadrons in Au+Au and U+U collisions with different $\beta_2$. The $\rho\left(v_2^2, [p_T]\right)$ shows strong non-trivial dependence on $\beta_2$. In particular, we observe the strongest sensitivity in the UCC region, where a large positive $\beta_2$ leads to a negative $\rho\left(v_2^2, [p_T]\right)$ due to the orientation of the colliding deformed nuclei. In the mid-central and peripheral regions, the magnitudes of $\rho\left(v_2^2, [p_T]\right)$ always decrease with increasing magnitude of $\beta_2$. The $\rho\left(v_3^2, [p_T]\right)$ are positive for both collision systems since triangular flow $v_3$ is purely fluctuation driven which is insensitive to the nuclear geometric effect. The detailed study can be found in Ref. [15].

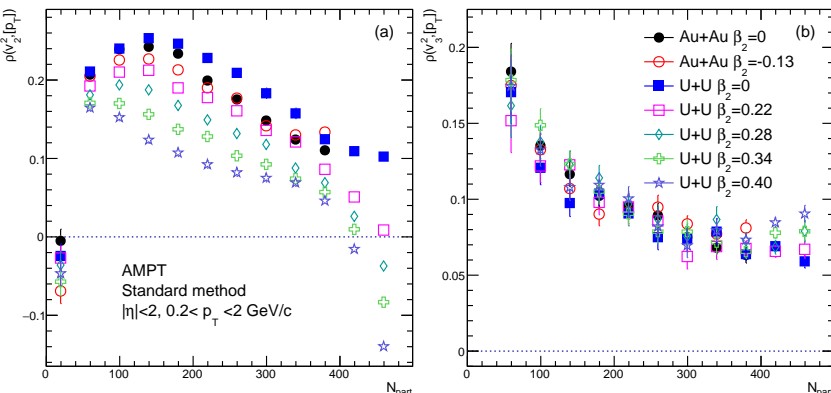

Figure 1: The $N_{part}$ dependence of the Pearson correlation coefficient $\rho\left(v_n^2, [p_T]\right)$ for n = 2 (left) and 3 (right) in Au+Au and U+U collisions with different deformation parameter $\beta_2$.

## 4 Conclusion

We studied the influence of the nuclear quadrupole deformation on the $\rho\left(v_n^2, [p_T]\right)$ in Au+Au and U+U collisions at RHIC energy using the AMPT transport model. $\rho\left(v_2^2, [p_T]\right)$ shows a strong dependence on $\beta_2$ and turns to change sign in the ultra-central collisions, while the $\rho\left(v_3^2, [p_T]\right)$ are always similar and positive. Detailed comparison of the model prediction with the results from STAR experimental data in Au+Au and U+U collisions could allow us to constrain the $\beta_2$ values of uranium nucleus.

## Acknowledgements

This work is supported by DOE DEFG0287ER40331 and NSF PHY-1913138.

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
