# Peer review of "Phenomenological study on correlation between flow harmonics and mean transverse momentum in nuclear collisions"

_SciPost Physics Proceedings, doi:SciPost Phys. Proc. 10, 009 (2022)_

## Round 3 · Referee Report · Anonymous (Referee 1) · 2022-1-27

Report

In this contribution to the ISMD2021 proceedings the authors summarize their study on the sensitivity of the Pearson coefficient, i.e. a correlator between elliptic flow and mean transverse momentum, in heavy ion collisions to deformations of the colliding nuclei. The manuscript is adequately written and meets the requirements.
I only have a few questions that the authors may want to address provided it is possible within the page limit:
1) Is it possible to define beta_2 or explain in a few words what it is?
2) The last sentence of section 2 (Analysis) about non-flow effects is difficult to understand for non-experts. Is it possible to comment in just a few words on the significance of non-flow contributions?
3) Which orientation of the nuclei gives rise to UCC events? Is it when the long axes line up on a single line?

  • validity: -
  • significance: -
  • originality: -
  • clarity: -
  • formatting: -
  • grammar: -

Author:  Chunjian Zhang  on 2022-02-04  [id 2160]

(in reply to Report 1 on 2022-01-27)

We would like to thank referee for kind efforts in reviewing our draft. In the new manuscript, we have explicitly expanded our emphasis on discussions, addressed the referee comments on the relevant parts.

We hope that our efforts and explanations could help you to arrive at a positive positon on the publication of our draft.

\begin{displayquote} {\color{blue}\noindent ---------------------------------------------------------------------- \ Report of Referee \ ---------------------------------------------------------------------- \ In this contribution to the ISMD2021 proceedings the authors summarize their study on the sensitivity of the Pearson coefficient, i.e. a correlator between elliptic flow and mean transverse momentum, in heavy ion collisions to deformations of the colliding nuclei. The manuscript is adequately written and meets the requirements.
} \end{displayquote}

We thank the referee for the positive assessment and useful suggestions. We have now made the changes in more compelling way in our new manuscript, added additional references. The revised manuscript is attached to this response. We hope that our case may now be considered enough to qualify for publication.

For more detailed comments, our replies to the raised questions are given below:

\begin{displayquote} {\color{blue}I only have a few questions that the authors may want to address provided it is possible within the page limit: 1) Is it possible to define $\beta_2$ or explain in a few words what it is?} \end{displayquote}

Thank referee for the insightful and useful comment. We defined it in our new manuscript as : \textit{$``$ where the $\beta_2$ values are obtained from the measured reduced electric transition probability $B(E_n)\uparrow$ via the standard formula $\beta_{2}=\frac{4 \pi}{3 Z R_{0}^{2}} \sqrt{\frac{B(E 2) \uparrow}{e^{2}}}$ [6]."}

\begin{displayquote} {\color{blue}2) The last sentence of section 2 (Analysis) about non-flow effects is difficult to understand for non-experts. Is it possible to comment in just a few words on the significance of non-flow contributions?} \end{displayquote}

Thank for the useful comment. We added more words for the contributions from nonflow as : \textit{$$The systematic study of short-range $$non-flow$"$ effect from resonance-decays, jets and dijets was checked in Ref. [11]."}

\begin{displayquote} {\color{blue}3) Which orientation of the nuclei gives rise to UCC events? Is it when the long axes line up on a single line?} \end{displayquote} Thank you for insightful question. For deformed nuclei collisions, both tip-tip and body-body collisions all could give rise to UCC events when the long axes line up on a single line. Based on the particle two-component model productions, tip-tip collisions with additional binary collisions ($\mathrm{N_{coll}}$) could contribute more ultra-central events. But if it's only based on the independent sources nucleon or quark Glauber particle production, both tip-tip and body-body contribute UCC equally.

We are hopeful that the clarifications given in this response may lead to a strong positive assessment of our manuscript.

---

## Round 4 · Author Response

1) Please find the above link to download the *.tex file in https://arxiv.org/format/2108.11452 2) Please use the newest version https://arxiv.org/pdf/2108.11452v4.pdf

---

## Round 4 · List of Changes

1) add the new definitions of beta_2 2) elaborate the nonflow source 3) add more relevant references.

---

## Editorial Decision

published